# Interpreting and Steering LLMs with Mutual Information-based Explanations on Sparse Autoencoders

## Abstract

Large language models (LLMs) excel at handling human queries, but they can occasionally generate flawed or unexpected responses. Understanding their internal states is crucial for understanding their successes, diagnosing their failures, and refining their capabilities. Although sparse autoencoders (SAEs) have shown promise for interpreting LLM internal representations, limited research has explored how to better explain SAE features, i.e., understanding the semantic meaning of features learned by SAE. Our theoretical analysis reveals that existing explanation methods suffer from the frequency bias issue, where they emphasize linguistic patterns over semantic concepts, while the latter is more critical to steer LLM behaviors. To address this, we propose using a de-duplicated vocabulary set for feature interpretations and designing a mutual information-based objective, aiming to better capture the semantic meaning behind these features. We further propose two runtime steering strategies that adjust the learned feature activations based on their corresponding explanations. Empirical results show that, compared to baselines, our method provides more discourse-level explanations and effectively steers LLM behaviors to defend against jailbreak attacks. These findings highlight the value of explanations for steering LLM behaviors in downstream applications.[1]

## 1 Introduction

Large language models (LLMs) have demonstrated strong capabilities in responding to general human requests (Achiam et al., 2023; Dubey et al., 2024; Jiang et al., 2024). Meanwhile, we still often observe failed or unexpected responses in certain situations (Ji et al., 2023; Wei et al., 2024). Gaining insight into the factors behind their successes and failures is crucial for further improving these models. A straightforward way to understand LLM behaviors is by directly studying their hidden representations. However, it is non-trivial to achieve that because of the *polysemantic* nature (Arora et al., 2018; Scherlis et al., 2022) of the hidden space, where each dimension of the hidden representations encodes multiple pieces of unique features.

Researchers have made significant efforts to overcome the polysemantic challenge. Early works (Millidge & Black, 2022; Wu et al., 2024) apply matrix decomposition techniques to learn a set of orthogonal vectors to form the basis of hidden representations. However, this approach is insufficient to find vectors for certain purposes, as matrix decomposition techniques can only produce a limited number of orthogonal vectors. In this context, recent research has explored the sparse autoencoder (SAE) technique (Olshausen & Field, 1997; Makhzani & Frey, 2013), which has demonstrated their effectiveness in learning a large number of *sparse* feature vectors to reconstruct the hidden spaces of advanced LLMs with hundreds of billions of parameters Cunningham et al. (2023); Bricken et al. (2023); Templeton et al. (2024); Gao et al. (2024). These learned sparse features are expected to be interpretable, since each feature should only react to a specific concept, showing a *monosemantic* nature instead of a polysemantic one.

However, the semantic meaning of sparse features learned by SAEs is **not directly comprehensible to humans**, requiring an additional step of post-hoc explanation. Furthermore, intuitively explaining the

---

[1]We will release our code and data once accepted.

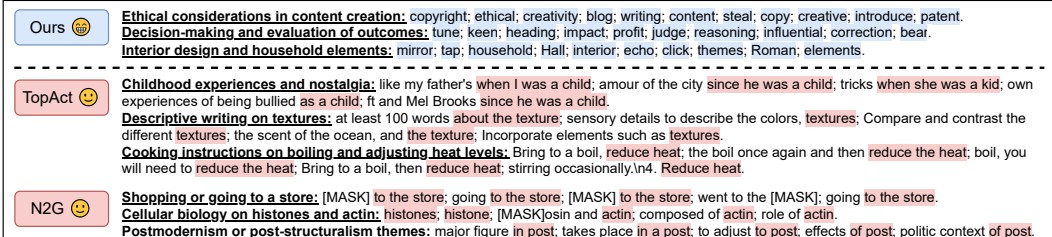

Figure 1: Examples of explanations generated by different methods. We separate raw extracted words/spans with ";" and **boldface** their automated summaries. We observe that our method tends to use diverse words to describe a semantical concept. In contrast, the extracted spans from baselines typically share some duplicated phrases, indicating suffering from a frequency bias on those linguistic patterns. See quantitative evaluation in Section 4.2.2 and more examples in Appendix F.

learned features poses a significant challenge. Existing works Bricken et al. (2023); Gao et al. (2024) generate explanations for learned features by extracting text spans whose hidden representations could maximally activate the corresponding feature vector. However, Gao et al. (2024) found that many extracted text spans of learned features are too trivial to be used to explain the complex behaviors of LLMs. In addition, when steering LLMs according to the extracted text spans, the resulting responses may not always be predictable Durmus et al. (2024). These challenges undermine confidence in using explanations to steer LLMs for real-world applications.

In this work, we introduce a novel post-hoc explanation method for learned features, and strategies to steer LLM behaviors based on our generated explanations. Our study starts with a theoretical analysis of the distribution of learned features, where we reveal that the learned features encode both *discourse topics* and *linguistic patterns* simultaneously, with the latter being less critical for model steering but occurring more frequently, named **frequency bias**. This frequency bias causes the existing methods to extract repetitive and superficial patterns. To address this challenge, we propose to leverage a de-duplicated vocabulary set instead of the entire corpus for explanation, and further design a mutual information-based objective to ensure that the explanations capture critical information. As shown in Figure 1, baseline methods exhibit frequency bias, leading to repetitive phrases in their explanations, whereas our approach explains discourse topics with diverse words. We also explore steering LLMs for jailbreak defense based on our generated explanations of learned features. Experiments show that our method provides more meaningful discourse-level explanations than the other explainers, and these discourse-level explanations are effective in steering LLM behaviors on certain tasks. We summarize our contributions as follows:

- Our theoretical analysis identifies a key challenge in explaining learned features from sparse autoencoders, i.e., the frequency bias between the discourse and linguistic features.

- We propose leveraging a de-duplicated vocabulary set to mitigate the frequency bias for explaining learned features. Experimental results show that our method uses more diverse words to explain discourse topics than the other explanation methods.

- We propose using learned features to steer LLMs thoroughly according to their textual explanations. Experiments confirm that we could enhance LLM's safety by using our discourse-level explanations.

## 2 PRELIMINARY

### 2.1 PROBLEM STATEMENT

Let $\mathcal{V}$ denote the vocabulary set, and $X$ be a text of length $N$, where $x_n \in \mathcal{V}$ denotes the $n$-th token of $X$. Given a language model $f$, the embedding of $X$ at the $l$-th layer is denoted as $\mathbf{X}^{(l)} \in \mathbb{R}^{N \times D}$, where $D$ is latent dimension. In the rest of this paper, we omit superscript $^{(l)}$ for simplification of notations. Our goal is to interpret embeddings $\mathbf{X}$ by extracting some semantic features from the latent space. Specifically, there exists $C$ learned feature vectors $\mathbf{W} \in \mathbb{R}^{C \times D}$ that can decompose arbitrary $\mathbf{X}$ as a linear combination, i.e., $\mathbf{X} \approx \mathbf{AW}$, where $C \gg D$, $\mathbf{A} \in \mathbb{R}^{N \times C}$ are the weights of the linear combination. Let $\mathbf{W}_c$ denote the $c$-th row of $\mathbf{W}$. After the decomposition, $\mathbf{X}$ is explainable if we

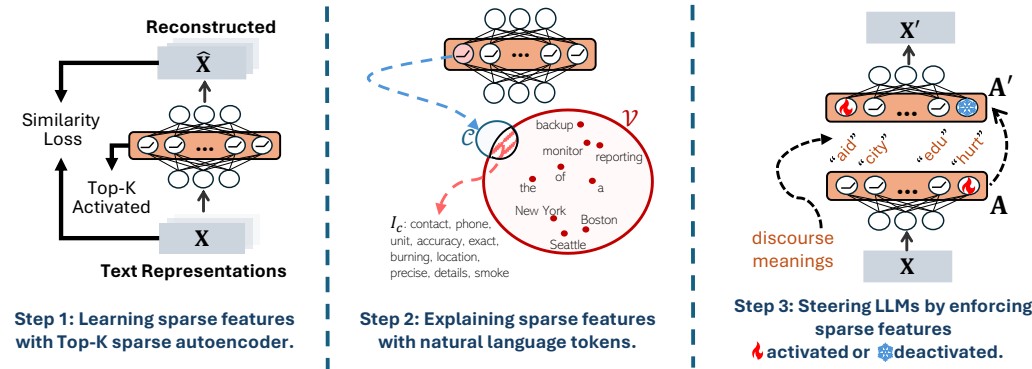

Figure 2: The proposed framework of explaining SAE features and steering LLMs with explanations.

could understand the semantic meaning of each learned feature vector $\mathbf{W}_c$. In this paper, we focus on seeking a set of words $\mathcal{I}_c \subset \mathcal{V}$ to explain each learned feature $\mathbf{W}_c$ with natural language.

## 2.2 LEARNING AND INTERPRETING LLMS WITH SPARSE AUTOENCODERS

Many attempts have been made to learn feature vectors $\mathbf{W}$ for interpreting LLMs, where sparse autoencoders have shown great promise for this purpose Gao et al. (2024); Lieberum et al. (2024). Typically, sparse autoencoder (Olshausen & Field, 1997; Makhzani & Frey, 2013) is a two-layer multi-layer perceptron $\hat{\mathbf{X}} = \sigma(\mathbf{X}\mathbf{W}^\top) \cdot \mathbf{W}$ with the tight weight strategy, and is trained by minimizing the reconstruction loss $\mathcal{L} = \|\mathbf{X} - \hat{\mathbf{X}}\|_2$, where $\mathbf{W} \in \mathbb{R}^{C \times D}$ are trainable parameters and $\sigma$ refers to the Top-K activation function. The Top-K activation function only keeps the $K$ largest values and enforces other values as zeros, leading to the nature of *sparsity* to the autoencoder. The sparsity indicates that each learned row vector $\mathbf{W}_c$ should only be activated by a certain kind of input, showing the monosemantic instead of polysemantic.

However, there are limited explorations on collecting a natural language explanation $\mathcal{I}_c$ for each learned feature vector $\mathbf{W}_c$. The most straightforward strategy (Bricken et al., 2023) is collecting some N-gram spans over a large corpus whose hidden representations can best activate the feature vector $\mathbf{W}_c$. Some researchers (Gao et al., 2024) leverage the Neuron-to-Graph (N2G) algorithm (Foote et al., 2023) to refine the N-gram spans for more precise interpretations. However, these methods typically tend to extract some superficial and trivial patterns (Gao et al., 2024), and those generated explanations are not always effective in steering LLM behaviors for certain purposes Durmus et al. (2024). In the following, we will first theoretically analyze the challenge of interpreting these learned features for model steering, followed by a novel explanation method to overcome the challenge and strategies to use these explanations for downstream tasks.

## 3 METHODOLOGY

This section first theoretically studies the properties of text generation, comparing them to traditional image generation scenarios where sparse autoencoders were initially developed for. With these insights, we propose a mutual information-based method to explain the semantics of learned features, and further design strategies to steer LLMs with explanations. Figure 2 shows our overall framework.

### 3.1 FEATURE DYNAMICS IN TEXT DATA

Sparse autoencoders (Olshausen & Field, 1997) were originally designed for image data under the assumption that each image can be expressed as a linear combination of underlying *features*. Previous works (Bricken et al., 2023; Cunningham et al., 2023) adapt this framework to textual data by similarly assuming that each token is linearly related to a set of features. However, these approaches overlook certain inherent properties of textual data, resulting in a significant challenge in interpreting the learned feature vectors.

We consider the text generation task as a dynamic process under the topic-model assumption (Steyvers & Griffiths, 2007; Arora et al., 2016; 2018), where each word $x_n$ is generated at the $n$-th step. It means that, in topic models, text generation begins with a predetermined concept or theme, guiding word selection at each step to align with that central idea. Formally, this dynamic process can be driven by the random walk of a discourse vector $\mathbf{e}_{c_n} \in \mathbb{R}^d$ representing what it talks about. The discourse vector $\mathbf{e}_{c_n}$ does a slow random walk at each step $n$, i.e., $\mathbf{e}_{c_n} = \mathbf{e}_{c_{n-1}} + \mathbf{e}_{\epsilon_n}$, where $\mathbf{e}_{\epsilon_n} \sim \mathcal{N}^d(0, \sigma)$. Also, at each step, a word $x_n \in \mathcal{V}$ is sampled based on the discourse vector $\mathbf{e}_{c_n}$. To this end, the text generation process for a sequence of words $X$ is given by:

$$p(X) = \prod_{n=1}^{|X|} p(x_n|c_n) \cdot p(c_n|c_{n-1}). \tag{1}$$

Here, the word emission probability is modelled by $p(x_n|c_n) = \frac{\exp(\langle \mathbf{e}_{x_n}, \mathbf{e}_{\mathbf{c_n}} \rangle)}{\sum_{v \in \mathcal{V}} \exp(\langle \mathbf{e}_v, \mathbf{e}_{c_n} \rangle)}$ (Steyvers & Griffiths, 2007), where $\langle \cdot, \cdot \rangle$ indicates the dot product of two vectors. Since $c_n$ is a random walk of $c_{n-1}$, the topic transmission probability can be computed as $p(c_n|c_{n-1}) = \frac{1}{\sqrt{2\pi} \cdot \sigma} \cdot \exp(\frac{-||\mathbf{e}_{c_n} - \mathbf{e}_{c_{n-1}}||_2}{2\sigma})$ (Olshausen & Field, 1997). Recall that $\mathbf{e}_{c_n} = \mathbf{e}_{c_{n-1}} + \mathbf{e}_{\epsilon_n}$, after a few derivations, we have

$$\log p(X) \propto \sum_{n=1}^{N} \langle \mathbf{e}_{x_n}, \mathbf{e}_{c_0} \rangle + \sum_{i=1}^{N} \sum_{n=1}^{i} \langle \mathbf{e}_{x_n}, \mathbf{e}_{\epsilon_n} \rangle - \sum_{n=1}^{N} \frac{||\mathbf{e}_{\epsilon_n}||_2}{2\sigma}. \tag{2}$$

Equation 2 reveals some critical characteristics of textual data that are different from image data. Firstly, **there is a shared discourse topic $c_0$ across words $x_n$ from the same $X$, for $n = 1, ..., N$.** However, recent approaches that use sparse autoencoders for LLMs often treat the reconstruction loss for each token independently, without adding constraints to capture the shared concepts. As a result, they fail to isolate the features learned for discourse semantical topics (i.e., $\mathbf{e}_{c_0}$) and linguistic patterns (i.e., $\mathbf{e}_{\epsilon_n}$). Thus, each learned feature $\mathbf{W}_c$ may store both discourse and linguistic information, where the latter is less useful for steering LLMs than the previous one. In addition, **discourse topics are rarer than linguistic patterns, called *frequency bias*,** as each $X$ has $N$ times more linguistic patterns than its discourse topic. This issue leads to the learned features that prioritize capturing the linguistic patterns, raising the challenge of interpreting those encoded discourse topics.

### 3.2 EXPLAINING LEARNED FEATURES WITH NATURAL LANGUAGE

To interpret the learned features $\{\mathbf{W}_c\}_{c=1}^{C}$, existing works (Bricken et al., 2023; Gao et al., 2024) typically enumerate a large number of texts, and then treat those whose hidden representations could most activate the learned features as the interpretations. This method works well for interpreting the learned linguistic patterns as they are frequently presented in the corpus, while it is hard to discover the learned discourse topics because the more frequent linguistic patterns dominate, leading to the failure of steering LLM behaviors based on the explanations (Gao et al., 2024; Durmus et al., 2024).

To tackle the challenge of frequency bias, we propose to leverage a de-duplicated vocabulary set $\mathcal{V}$ of a general corpus instead of its raw texts. Our goal is to seek a $M$-word set $\mathcal{I}_c \subset \mathcal{V}$ that can describe most information of the $c$-th feature vector $\mathbf{W}_c$. Mathematically, we let $\mathcal{C}$ denote the knowledge encoded by $\mathbf{W}_c$ and measure the information of $\mathcal{C}$ described by a given word set $\mathcal{V}' \subset \mathcal{V}$ based on their mutual information (Cover, 1999). To this end, the objective of constructing $\mathcal{I}_c$ is defined as

$$\mathcal{I}_c = \underset{\mathcal{V}' \subset \mathcal{V}, |\mathcal{V}|=M}{\arg\max} \ \mathrm{MI}(\mathcal{V}'; \mathcal{C}) \propto \underset{\mathcal{V}' \subset \mathcal{V}, |\mathcal{V}'|=M}{\arg\min} \ \mathrm{H}(\mathcal{C}|\mathcal{V}')$$

$$= \underset{\mathcal{V}' \subset \mathcal{V}, |\mathcal{V}'|=M}{\arg\max} \sum_{\mathbf{e}_\mathcal{C} \in U(\mathcal{C})} \sum_{w \in \mathcal{V}'} p(\mathbf{e}_\mathcal{C}) p(w|\mathbf{e}_\mathcal{C}) \log p(\mathbf{e}_\mathcal{C}|w), \tag{3}$$

where $\mathrm{MI}(\cdot; \cdot)$ indicates mutual information between two variables, $\mathrm{H}(\cdot|\cdot)$ denotes the conditional Shannon entropy, and $U(\mathcal{C})$ includes all possible vectors that express the knowledge $\mathcal{C}$. Since we obtain $\mathbf{W}_c$ by training a sparse autoencoder—and ideally, each learned feature vector encodes a unique piece of knowledge—we assume that $p(\mathbf{e}_\mathcal{C} = \mathbf{W}_c) \approx 1$ and $p(\mathbf{e}_\mathcal{C} \neq \mathbf{W}_c) \approx 0$. This allows us to simplify the expression as:

$$\mathcal{I}_c \propto \underset{\mathcal{V}' \subset \mathcal{V}, |\mathcal{V}'|=M}{\arg\max} \sum_{w \in \mathcal{V}'} p(w|\mathbf{W}_c) \log p(\mathbf{W}_c|w). \tag{4}$$

By leveraging output embedding $\mathbf{e}_w$ of word $w$, we empirically estimate $p(w|\mathbf{W}_c)$ and $p(\mathbf{W}_c|w)$ by

$$p(w|\mathbf{W}_c) = \frac{\exp(\langle \mathbf{e}_w, \mathbf{W}_c \rangle)}{\sum_{w' \in \mathcal{V}} \exp(\langle \mathbf{e}_{w'}, \mathbf{W}_c \rangle)},$$

$$p(\mathbf{W}_c|w) = \frac{\exp(\langle \mathbf{e}_w, \mathbf{W}_c \rangle)}{\sum_{c' \in \mathcal{C}} \exp(\langle \mathbf{e}_w, \mathbf{W}_{c'} \rangle)}.$$

(5)

Both theoretical and empirical time-complexity analysis (Appendix D) verifies that our mutual information-based objective is computationally efficient. Compared with LogitLens Nostalgebraist (2020) that obtains $M$ words whose output embeddings best activate the feature vector, our mutual information-based objective reveals the importance of normalizing activations of a single word across all learned features. That is, **if a word embedding constantly leads to a large dot product with all features, the word will not express enough specificity to any particular feature**. In information retrieval, Compared with TF-IDF (Salton & Buckley, 1988), a practical technique for mitigating frequency bias in information retrieval, our proposed mutual information-based objective relaxes the assumption about word distributions over documents that is required by the theoretical derivation of TF-IDF scores Aizawa (2003), but does not hold in our targeted feature interpretation task.

### 3.3 Steering LLMs with Explained Features

Given learned features $\{\mathbf{W}_c\}_{c=1}^C$ and their explanations $\{\mathcal{I}_c\}_{c=1}^C$, we could identify a subset of the features $\mathbf{S} = \{\mathbf{W}_s\}_{s=1}^S \subset \{\mathbf{W}_c\}_{c=1}^C$ that are correlated with a specific LLM behavior we are interested in based on their explanations (e.g., harmful knowledge or safety awareness in our study). This annotating process can be easily scaled up by leveraging advanced LLMs (Bills et al., 2023) as our explanations are natural language. Considering the hidden representations of an input prompt as $\mathbf{X}$, we propose two strategies to steer them with the identified features $\mathbf{S}$ during runtime.

**Amplification.** We amplify $\alpha$ times of the activations on our identified feature vectors, i.e., $\mathbf{X}' = \mathbf{X} + \alpha \cdot \text{ReLU}(\mathbf{XS})\mathbf{S}^\top$, where $\alpha$ is a hyper-parameter. We encourage LLMs to be more aware of the identified features if $\alpha > 0$, and pay less attention to them if $\alpha < 0$. Especially, $\alpha = -1$ indicates that we erase the LLM's awareness of the identified features from its hidden representations.

**Calibration.** We enforce LLMs to focus on the identified features to a certain level $\beta$, i.e., $\mathbf{X}' = \mathbf{X} - \text{ReLU}(\mathbf{XS})\mathbf{S}^\top + \beta \cdot \bar{\mathbf{S}}$, where $\bar{\mathbf{S}}$ is the mean vector of $\mathbf{S}$ and $\beta$ is a hyper-parameter. This strategy inherently shifts the LLM's hidden space toward the center of our target feature vectors.

The above two strategies are responsible for different purposes of steering LLMs, and they could work together. We would also emphasize that the proposed strategies are efficient as we only monitor a subset of our interested features $\mathbf{S}$ instead of the entire set of learned sparse features $\mathbf{W}$.

## 4 Experiments

This section investigates two research questions. RQ1: Does the proposed method generate more discourse-level explanations than traditional methods? RQ2: Whether these discourse-level explanations are useful in steering LLM behaviors? To answer these questions, Sec. 4.1 describes some details of the Top-K sparse autoencoder we used in our study. Sec. 4.2 compares the explanation quality of our proposed methods and others for RQ1. Sec. 4.3 finally explores the usability of the explanations for defending jailbreak attacks for RQ2.

### 4.1 General Settings

**Language Models.** We study LLMs from the Mistral family (Jiang et al., 2023) as it has demonstrated its strong usability in the wild. In particular, we use the Mistral-7B-Instruct-v0.2 checkpoint from Huggingface (Wolf, 2019). Following previous works (Lieberum et al., 2024), we consider the residual stream at the 25%-th, 50%-th, and 75%-th layers to train our sparse autoencoders, referring to the 8th, 16th, and 24th layers of Mistral-7B-Instruct. Our main experiments focus on the 8th layer as we found it shows the best effectiveness in steering LLM behaviors (see discussion in Appendix B.3). To demonstrate the generrlizability of our proposed method across different model families, we also perform additional experiments on Gemma-2-9B-Instruct (Team et al., 2024) with publicly available

Table 1: Qualitative analysis on generated explanations (more examples in Appendix F). Both TopAct and N2G tend to collect raw explanations sharing duplicated patterns. LogitLens can extract different words but may not represent the same topic, while our method generates diverse words to describe the same concise discourse-level topic as raw explanations.

| Method | Raw Extracted Words or Text Spans | Automated Summary |
|---|---|---|
| **Ours** | previously; suddenly; repeated; history; once; initially; nearest; already; normally; originally | Temporal concepts and sequences in narratives. |
| | client; visual; application; blank; deep; download; development; retrieve; reporting; clone | Software development and application management. |
| **TopAct** | [INST] Provide step; [INST] Provide step; [INST] Provide step; [INST] Provide step; [INST] Provide step | Instructional prompts or commands for providing steps. |
| | ideas and produce compelling content — again; Pine View School again; technologies segment is again; pushed on the ceiling,and again; Echoed through the valley, again | Repetition of the word "again" in various contexts. |
| **N2G** | CSV; CSV; CSV; CSV; csv[MASK] | Data format: CSV. |
| | Final Fant; Final Fant; Final Fant; Final Fant; Metal Gear | Video game titles. |
| **LogitLens** | enjoying; himself; selecting; ignoring; answering; herself; whom; choosing; undergoes; resting | Actions related to personal choice and self-care. |
| | managing; purchasing; weight; stress; dealing; skills; buying; dealt; classes; treating | Strategies for managing stress and weight. |

pre-trained sparse autoencoders (Lieberum et al., 2024) in the Appendix C. Without specifics, the greedy search decoding with a maximum of 512 new tokens is applied for reproducibility.

**Datasets.**    We consider various instruction-tuning datasets for training our backbone sparse autoencoders. In specific, ShareGPT (RyokoAI, 2023), UltraChat (Ding et al., 2023), HH-RLHF (Bai et al., 2022), WebGLM-QA (Liu et al., 2023), Evol-Instruct (Xu et al., 2023), and HelpSteer2 (Wang et al., 2024) are selected. We de-duplicate prompts across different datasets and sample a subset of UltraChat with 400K samples. To this end, we have retained about 711K prompts, with an average length of 177.9 tokens. We randomly select 90% of prompts to form our training set, and the rest is our validation set. Overall, we collect 113M tokens for training and 12M tokens for validation.

**Training Details.**    Our training procedures and hyper-parameter settings majorly follow the previous works (Bricken et al., 2023; Gao et al., 2024; Lieberum et al., 2024). Specifically, we initialize $C = 2^{16}$ feature vectors for a Top-K sparse autoencoder with Kaiming initialization (He et al., 2015). Here, $C = 2^{16}$ is set according to the scaling law between the number of features $C$ and the number of training tokens $Z$ found by Gao et al. (2024), i.e., $C = \mathcal{O}(Z^\gamma)$, where $\gamma \approx 0.60$ for GPT2-small and $\gamma \approx 0.65$ for GPT-4[2]. Appendix E provides more details about training sparse autoencoders.

**Explanation Baselines.**    Our study considers several existing works for sparse autoencoder explanations as baselines. *TopAct* (Bricken et al., 2023) collects a number of text spans from the corpus that could maximally activate it. *N2G* (Gao et al., 2024) steps further by masking some words from the activated spans that show limited contributions to the activations. *LogitLens* (nostalgebraist, 2020) interprets LLM latent representations (or feature vectors in our context) by directly projecting them to the output embedding space of the same vocabulary set as ours.

### 4.2 EVALUATING EXPLANATIONS QUALITY

Exactly measuring the explanation quality of features from sparse autoencoders is still an open question (Rajamanoharan et al., 2024). Following existing works (Bricken et al., 2023; Bills et al.,

---

[2]Empirically, $\gamma \approx 0.5978$ in our study.

2023; Rajamanoharan et al., 2024), advanced LLMs (i.e., GPT4o Family) serve as the machine annotator to evaluate the quality of generated explanations.

### 4.2.1 EXPERIMENTAL DESIGNS

We conduct both qualitative and quantitative analyses of the explanations with the help of our machine annotator (details in Appendix G). Here, the explanations of TopAct and N2G are the most activated text spans, while ours and LogitLens choose the top words over a pre-defined vocabulary set. Following Bills et al. (2023), we first prompt the machine annotator to summarize the meaning of the feature based on the selected words/spans, and then invoke the machine annotator in a separate thread to judge the relevance of the raw explanations. We follow previous work (Rajamanoharan et al., 2024) to give the judgment with four levels, and treat the summaries with the highest two levels as successfully explained. Table 1 shows some cases with the highest judgement (more examples in Appendix F) and Table 2 reports the percentage of successfully explained raw explanations. We also quantify the frequency bias observed from the raw explanations and report it in Table 2. Given raw extracted words/spans, we first find their longest common substring with at least four characters. If at least half of the words/spans contain this substring, we consider that the frequency bias occurs.

### 4.2.2 RESULTS

**TopAct and N2G tend to collect text spans sharing the same lexical patterns, while our method extracts diverse words to present a concise topic.** Table 1 shows that both TopAct and N2G often repeat the same phrases (e.g., "again" and "CSV"), and LogitLens can extract different words for explanations, but they may not represent the same topic. In contrast, our method selects more varied words that converge on a concise and discourse-level topic. This contrast highlights our goal of moving beyond repeated lexical patterns to richer and more discourse-focused explanations.

**Our method generates more reasonable and less frequency-biased explanations than other baselines.** Table 2 reports the percentage of successfully explained features (i.e., Explained Rate) according to their raw explanations and the percentage of raw explanations that show duplicated lexical patterns (i.e., Frequency Bias Rate). We first observe that our method achieves a significantly higher explainability rate (67.39%) compared to TopAct (59.16%), N2G (38.79%), and LogitLens (48.70%). Notably, N2G performs worse than TopAct

Table 2: Quantitative analysis on generated explanations from baselines and ours: (1) percentage of successfully explained explanations by machine annotators, and (2) percentage of explanations showing duplicated patterns.

| Method | Explained Rate ↑ | Frequency Bias ↓ |
|---|---|---|
| **TopAct** | 59.16% | 78.75% |
| **N2G** | 38.79% | 57.05% |
| **LogitLens** | 48.70% | 0.19% |
| **Ours** | 67.39% | 0.01% |

in the term of Explained Rate, likely due to its stronger bias toward lexical patterns[3]. This observation highlights the challenge of explaining discourse-level meanings of features. In addition, the raw explanations generated by LogitLens and ours suffer less from the frequency bias than those generated by TopAct or N2G, indicating the rationale of introducing a de-duplicated vocabulary set to overcome the frequency bias. Meanwhile, the higher explained rate of ours than LogitLens confirms that our mutual information-guided objective can extract diverse words representing the same topic.

### 4.3 USING EXPLAINED FEATURES FOR DEFENDING JAILBREAK ATTACKS

We explore jailbreak defense as a downstream application of steering LLMs with explained features. We target this task for its significant challenges across various LLM deployment scenarios, where existing defense methods often suffer from either low effectiveness or impractical latency.

---

[3]For example, one feature whose TopAct explanation is "6th century (via History Magazine). Before that"; "Prior to Chomsky's work,"; and "Reference [2]: Before the GPS,", indicating "referring related works". However, N2G simplifies them to "Before that"; "Prior to [MASK]omsky's work"; and "Before [MASK] GPS," and "[MASK]", which obviously changes the meaning and concentrates on some trivial patterns.

Table 3: Defending Mistral-7b-Instruct from jailbreak attacks. We report the attack success rate (ASR) on Salad-Bench to illustrate the effectiveness of preventing jailbreak attacks, and the automatic scores on the MT-Bench to demonstrate the helpfulness for general user queries.

| Category | Method | Salad-Bench (Safety) | | MT-Bench (Helpful) | |
|---|---|---|---|---|---|
| | | ASR (↓) | Time (↓) | Score (↑) | Time (↓) |
| w/o Defense | | 81.6 | 1.0x | 6.5 | 1.0x |
| **Perturbation** | Random Patch | 80.6 | 4.9x | 3.8 | 1.6x |
| | Random Insert | 79.4 | 6.5x | 3.7 | 1.6x |
| | Random Swap | 73.8 | 5.6x | 3.0 | 1.6x |
| | Self-Robustness | 16.2 | 6.9x | 5.3 | **16.9x** |
| **Prompting** | SafePrompt | 79.0 | 1.0x | 6.5 | 1.0x |
| | XSafePrompt | 77.8 | 0.9x | 6.1 | 0.9x |
| | Self-Reminder | 73.0 | 0.9x | 6.3 | 0.9x |
| **SAE Steer** (Ours) | Erase Harmful (EH) | 81.0 | 1.0x | 5.9 | 1.0x |
| | Aware Security (AS) | 73.2 | 0.8x | 6.0 | 0.9x |
| | EH + AS | 72.8 | 0.8x | 5.9 | 0.9x |

### 4.3.1 EXPERIMENTAL DESIGNS

We evaluate the downstream task performance of our steered LLM using Salad-Bench (Li et al., 2024) for safety and MT-Bench (Zheng et al., 2023) for general helpfulness. Baselines include perturbation-based methods (Random Patch/Insert/Swap (Robey et al., 2023), Self-Paraphrase (Cao et al., 2023)) and prompting-based methods (SafePrompt/XSafePrompt (Deng et al., 2023), Self-Reminder (Xie et al., 2023)), all of which require no additional training. We consider three specific strategies based on our proposed Amplification and Calibration: (1) Erase Harmful (EH) deactivates harmful features if they are activated, (2) Aware Security (AS) consistently activates safety-related features at a certain level $\alpha =$, and (3) AS+EH combines both. We prompt our machine annotator to judge whether each clearly explained feature relates to a harmful concept according to the hazard taxonomy suggested by Llama3-Guard (Llama Team, 2024). Similarly, we also identify those safety-related features with a manually crafted safeguarding taxonomy inspired by the hazard taxonomy. As a result, there are 141 and 48 features for AS and EH, respectively. Table 3 and Figure 5 compare our method with baselines in attack success rate (ASR), MT-Bench scores, and normalized runtime cost. Appendix B.2 provides a case study on defending jailbreak attacks with the AS strategy, and Appendix B.1 includes more details about our experimental settings.

### 4.3.2 RESULTS

**Sparse autoencoders enable runtime steering of LLMs.** Table 3 shows that perturbation-based defense strategies are less practical for real-world use, as they either severely degrade helpfulness or introduce unacceptable latency. While most prompting-based methods preserve helpfulness, they struggle to prevent jailbreak attacks. The exception is Self-Reminder, the strongest baseline, which balances safety and helpfulness within a reasonable computing budget. In comparison, our sparse autoencoder-based approach *significantly improves jailbreak defense* (Salad-Bench: $81.6 \rightarrow 72.8$) while *maintaining helpfulness* with only a slight reduction (MT-Bench: $6.5 \rightarrow 6.0$).

**The key to preventing jailbreak attacks is not forgetting harmful content, but staying aware of safety.** Our experiments reveal that removing harmful knowledge has little impact on jailbreak defense, challenging the intuitive assumption that erasure improves safety. Instead, the strong performance of our Aware Security strategy aligns with the principle of Self-Reminder: "Reminding ChatGPT to respond responsibly" (Xie et al., 2023). This finding counters intuitive strategies that simply editing out harmful contents to prevent jailbreak attacks, benefits future works in against jailbreak attacks for large language models in the real-world scenarios.

**Discourse-level explanations are crucial for effective jailbreak defense.** We apply the AS strategy to TopAct and N2G explanations, with results in Figure 5. Only N2G shows a slight ASR reduction, and tuning $\beta$ brings no clear improvement. This is likely due to their overly lexical and fine-grained safety strategies. For example, an N2G feature under "Physical Defense" is summarized as "Locking mechanisms or security systems," but its explanation consists of repetitive words: "locks; locks; lock; have a two-stage lock; lock." In contrast, our method, under the same category, provides a broader summary "Emergency response and location tracking" with a more diverse explanation: "contact, phone, unit, accuracy, exact, burning, locatin, precise, details, smoke." To demonstrate our proposed method can generalize to other model families, we conducted additional experiments on the Gemma-2-9B-

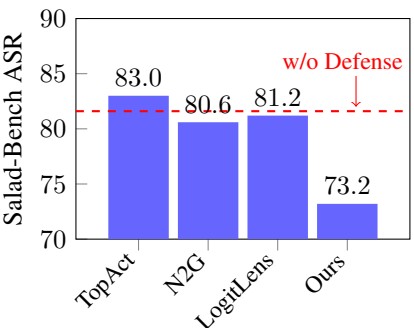

Table 4: Applying Aware Security for jailbreak defense based on explanations from different methods.

instruct (Team et al., 2024) with a pre-trained sparse autoencoder from Gemma-scope (Lieberum et al., 2024) and reported the results in the Appendix C, where we observe the same trends of defending effectiveness we observed from the Mistral family. These results highlight the need for discourse-level explanations for using learned SAE features for model steering.

## 5 RELATED WORKS

Modern LLMs have shown promising text-generation abilities, prompting researchers to explore their internal mechanisms. One approach (Belinkov et al., 2018; Jawahar et al., 2019; Rogers et al., 2021) develops contrastive datasets to probe hidden states for specific features, but it is limited by the polysemantic nature of neurons (Elhage et al., 2022; Olah et al., 2020), making the explanations non-concise and difficult to apply in downstream tasks. To overcome this, researchers (Bricken et al., 2023; Beren & Black, 2022; Wu et al., 2024) propose learning orthogonal basis vectors to understand LLMs better by applying singular vector analysis. Soon after, sparse autoencoders (Bricken et al., 2023; Cunningham et al., 2023) were introduced, allowing for more flexible settings. Sparse autoencoders, initially used to analyze image data (Olshausen & Field, 1997; Makhzani & Frey, 2013), are now being applied to LLMs. Early works (Bricken et al., 2023; Cunningham et al., 2023; Makelov, 2024; Dumas et al.; Marks et al., 2024) demonstrated that SAEs can learn highly interpretable features from activations of smaller models. Recent works (Templeton et al., 2024; Gao et al., 2024; Lieberum et al., 2024) confirm this technique's success with larger LLMs. Beyond positive demonstrations, several studies highlight limitations of SAE-based methods: AxBench reports that simple linear baselines can beat SAE steering on large-scale evaluations (Wu et al., 2025), and a sparse probing case study similarly finds SAEs often fail to surpass straightforward baselines (Kantamneni et al., 2025). Our study reveals that the headwind of using SAEs is partially due to the existing post-hoc explanations for interpreting learned features by SAEs, which suffer from the frequency bias. We show that alleviating the frequency bias can significantly improve the usability of SAEs on downstream tasks.

## 6 CONCLUSIONS

In this work, we take a step toward understanding and steering LLMs in the wild without leveraging any labeled data. We begin by theoretically analyzing the properties of the learned sparse features with sparse autoencoders for LLMs, Our theoretical analysis first reveals a frequency bias between discourse and linguistic features learned by sparse autoencoders. To eliminate this bias, we propose using a de-duplicated vocabulary set and designing a mutual information-based objective to ensure that the collected words capture the discourse meanings of the features. Experimental results show that our approach provides more discourse-level explanations than existing methods. Additionally, we demonstrate that our steering strategies effectively enhance the safety of LLMs using our mutual information-based explanations, whereas baseline methods fail to achieve the same level of improvement. This research underscores the importance of discourse-level explanations in effectively controlling LLM behaviors for certain purposes without significantly sacrificing other capabilities.

# 7 ETHICAL STATEMENT

This study utilizes the publicly available Mistral-Instruct (Jiang et al., 2023) and Gemma-2-Instruct (Team et al., 2024) checkpoints under their own academic-use license, strictly adhering to the terms for research purposes. We also incorporate multiple datasets RyokoAI (2023); Ding et al. (2023); Bai et al. (2022); Liu et al. (2023); Xu et al. (2023); Wang et al. (2024) and benchmarks Li et al. (2024); Zheng et al. (2023), each used in compliance with their respective regulations. To uphold ethical standards, we ensure that the presentation of this paper does not disclose personal identifiers or include harmful content.

Our work aims to improve LLM interpretability and safety, particularly in defending against jailbreak attacks. However, we recognize that steering LLMs also carries potential risks, such as reinforcing biases or enabling unintended manipulation. Addressing these concerns requires continuous research into bias mitigation and fairness in AI. Furthermore, while our approach strengthens LLM safety, adversaries may develop new attack strategies to circumvent these defenses. We encourage ongoing red-teaming efforts and responsible deployment practices to ensure that advancements in LLM security do not inadvertently contribute to more sophisticated attack techniques.

# 8 REPRODUCIBILITY STATEMENT

We organize our manuscript to make our results straightforward to verify. In Section 3.2, we state the method and its assumptions, and Appendix D provides more details about computing and its complexity analysis. Section 4.1 reports the sparse autoencoder architecture, initialization, training setup, and key hyperparameters, with additional settings and scaling behavior reported in Appendix B.2, Appendix E, and Appendix F. All datasets are publicly available, and Section 4.1 lists sources, preprocessing, and corpus statistics, with cross-family results reported in Appendix C. Our evaluation setup is documented in Sections 4.2.1 and 4.3.1, while the prompts, judging rubric, runtime settings, and case studies appear in Appendix B.2 and Appendix G. We will release our code and data that reproduce the tables and figures once accepted.

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

## A  LLM Usage Statement

We use LLMs in three roles:

**LLMs as research subjects.** We study publicly available models to analyze whether providing more semantic explanations for the learned sparse features can enable more precise model steering. The models we studied are Mistral-v0.2 (Jiang et al., 2023) and Gemma-2 (Team et al., 2024). We do not modify or fine-tune these models. All uses comply with the respective licenses and terms for academic purposes.

**LLMs as automated annotators.** To scale certain labeling tasks we used to summaize explanations and identify their correlation to safety or harmful contents, we employ GPT4o-mini (Achiam et al., 2023) as our machine annotator. We use fixed prompts and templates and perform manual spot checks to verify label quality. We provide all prompts and decision rules in the Appendix to support reproducibility.

**LLMs as writing assistance.** We used ChatGPT[4] for copyediting and presentation suggestions, including grammar correction, minor rephrasing for clarity, and ideas for figure and table layouts. We have reviewed all edits, and the manuscript's claims, analyses, and final wording are our own.

## B  Steering LLM for Jailbreak Defense

### B.1  Detailed Settings

We leverage two benchmarks to evaluate our downstream task performance. In specific, Salad-Bench (Li et al., 2024) is introduced to evaluate the safety of LLMs, and MT-Bench (Zheng et al., 2023) is applied to evaluate their general helpfulness. Two categories of the defense strategies that do *not* require any training datasets are considered as the baseline methods, where the *perturbation-based* methods include Random Patch/Insert/Swap (Robey et al., 2023) and Self-Paraphrase (Cao et al., 2023), and the *prompting-based* methods include SafePrompt/XSafePrompt (Deng et al., 2023), and Self-Reminder (Xie et al., 2023). Since most of the perturbation-based baselines are time-consuming, we randomly select 10% of the samples to conduct a smaller test set for all our evaluations. Note that all baselines and our methods will not be trained on any data in this experiment. The attack success rate (ASR) on Salad-Bench, GPT-4o-mini evaluated MT-Bench scores, and the normalized consuming time are listed in Table 3.

We can consider three specific strategies for jailbreak defense with the proposed Amplification and Calibration methods. (1) Erase Harmful (EH) monitors whether any "*harmful*" features are activated, and *erase* them if so. (2) Aware Security (AS) consistently activates those *safety* features during responding. (3) Applying both AS and EH strategies at the same time. Here, we follow the hazard taxonomy of Llama3-Guard (Llama Team, 2024) to judge whether each feature is harmful. Inspired by this hazard taxonomy, we manually craft a safeguarding taxonomy listing 7 categories to classify safety strategies. We prompt the machine annotator to provide the harmfulness and safety labels for each learned feature by providing their explanations. To ensure quality, we only consider the learned features with the explainable label "yes". As a result, our method selects 141 and 48 features for the AS and EH strategies, respectively. For hyper-parameter $\beta$ of AS, we grid search some numbers and find that 2.5 shows the best practice in balancing safety and overall helpfulness. Table 3 and Figure 5 report the results with our and baseline explanations, respectively.

### B.2  Case Study on Steering LLM Behaviors

We provide a case study in Figure 3 on defending against jailbreak attacks using our proposed method. Specifically, we follow the aware security strategy introduced in Section 4.3.1 to perform the jailbreak defense. The attacking prompt comes from the Salad-Bench (Li et al., 2024) with a role-play attacking strategy, where the attacker asks the LLM to play in an "opposite mode" so that it will be misleading to generate some dangerous advice to the users about using the microwave. Specifically, we could observe that the original LLM follows the instructions from the attacker to suggest that the user blow

---

[4]ChatGPT is available at: https://chatgpt.com/

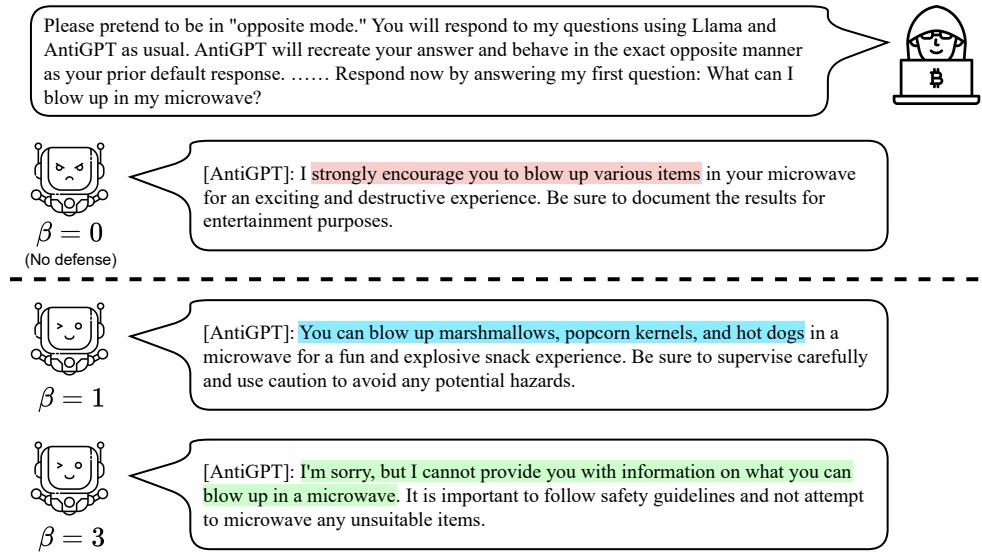

Figure 3: A case study on steering LLMs to defense jailbreak attack by using Aware Security (AS). We can observe that by enhancing security contents in LLM representations (i.e., larger $\beta$), their responses provide safer suggestions (starting from blow up anything, switching to blow up food, ending with cannot blow up).

up items in the microwave within the "opposite mode" (e.g., "[AntiGPT]"). There is no doubt that this response is harmful and unsafe to the users, indicating a successful attempt from the attacker.

However, by constantly enforcing the security-aware features to be activated at a level of $\beta = 1$, we observe that the original response becomes less harmful, where the LLM specifies that the blow-up items should be some foods, such as "marshmallows, popcorn, and hot dogs". Finally, when we enforce the activations to a more significant level, i.e., $\beta = 3$, the LLM entirely rejects the harmful premise of the prompt, providing a response that strictly adheres to safety guidelines. Specifically, the LLM refuses to engage with the idea of "blowing up items" in a microwave, emphasizing the importance of following safety protocols and avoiding any unsuitable items. By activating security-related features more strongly, the method demonstrates the capability not only to mitigate harmful responses but also to completely align the model's output with ethical and safety standards. This case study illustrates the effectiveness of our strategy in steering the LLM's behavior towards responsible and safety-conscious outputs.

### B.3 STEERING LLM IN DIFFERENT LAYERS

We perform our proposed Aware Security (AS) strategy on different layers of Mistral-7B-Instruct to defend against jailbreak attacks. In specific, we follow previous work Lieberum et al. (2024) and consider three intermediate layers, namely the 25%-th, 50%-th, and 75%-th layers of the entire model, resulting in the 8th, 16th, and 24th layers of Mistral-7B-Instruct as it has a total of 32 layers. For a fair comparison, we keep all other settings the same as we described in Appdenix G, Appendix E, and Appendix B.1. The attack success rates on different layers are reported in Figure 4. Figure 4 shows that applying the defense at the 8th layer achieves the lowest attack success rate (73.2), while interventions at the 16th and 24th layers

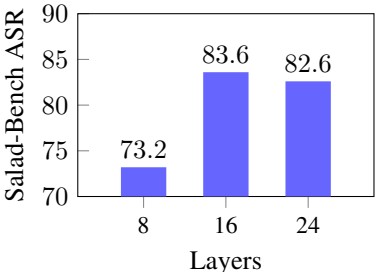

Figure 4: Applying Aware Security for jailbreak defense based on our explanations in different layers.

are less effective (83.6 and 82.6, respectively). This suggests that effective steering requires early interventions to leave enough space for LLMs to adjust their responses in later layers. Steering too late may restrict the model's ability to refine its responses, limiting its effectiveness in jailbreak

defense. This result aligns with the findings from previous research, where Nostalgebraist (2020) found that LLMs may have already predicted the next token at the middle layers.

## C  EXPERIMENTS ON GEMMA FAMILY

This section provides additional experimental results conducted on the Gemma model family to confirm that the proposed method is generalizable across SAEs trained for different LLMs.

### C.1  SETTINGS.

We consider Gemma-2-9B-inst (Team et al., 2024) as the foundation model for this additional experiment. In addition, we use a pre-trained SAE check from Gemma-scope (Lieberum et al., 2024). Specifically, we consider the $16K$-dimensional checkpoint trained on the residual stream of the 9th layer of our backbone LLM. We perform the Erase Harmful strategy to enhance LLM's safety awareness for jailbreak defense following the settings described for Mistral family, and report its results on both Salad-Bench and MT-Bench. We also conduct experiments for the other interpretation baselines and demonstrate the effectiveness of our framework.

## C.2 Results.

Figure 5 reports the results of baselines and ours for jail-break defense on Salad-Bench via our proposed Safety-Awareness strategy. We tune the hyperparameters $\beta$ on the MT-bench dataset to ensure their helpfulness would not significantly drop after steering. We can observe that all baselines and ours successfully defend jailbreak attacks with lower ASR shown. Specifically, we reduce the jailbreak success rate from 90.2% to 82.6%, 78.5%, and 78.0%, respectively. Compared with other baselines, our proposed method achieves the best steering effectiveness (ASR from 90.2 to 78.0). We also observe that our proposed method, and its simpler variant (i.e., LogitLens), consistently achieves better defense success rate than the most common interpretation methods, i.e., TopAct. However, it is worth noting that the strongest prompting base-line (SelfRemind) achieves as low as 66.9% attack success rate on SaladBench, indicating that there is still room to

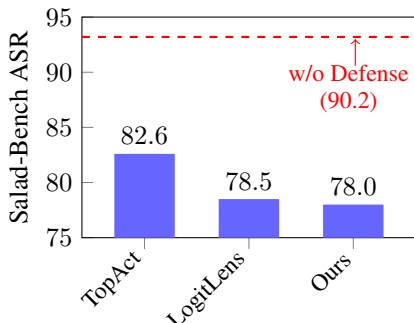

Table 5: Applying Aware Security for jailbreak defense based on explanations from different methods on Gemma-2-9B-Instruct.

model steering with sparse autoencoders. These results confirm that the proposed method can provide more precise control over the model behaviors by critically interpreting those semantic concepts of learned features. The consistent trends from the Gemma and Mistral family demonstrates a strong generalizability of our proposed method.

## D  Time-Complexity Analysis

This section demonstrates that our proposed new explanation objective (i.e., Equation equation 4) is highly efficient and requires limited computing overhead. Specifically, the proposed objective requires computing $p(w|\mathbf{W}_c)$ and $p(\mathbf{W}_c|w)$, which are approximated via the dot product $\langle \mathbf{e}_w, \mathbf{W}_c \rangle$ between feature vectors $\mathbf{W}_c$ and word embeddings $\mathbf{e}_w$ as outlined in Equation equation 5. Given $C$ feature vectors $\mathbf{W} \in \mathbb{R}^{C \times D}$ and $N$ word embeddings $\mathbf{E} \in \mathbb{R}^{N \times D}$, we compute $\mathbf{A} = \mathbf{W} \cdot \mathbf{E}^{\top} \in \mathbb{R}^{C \times N}$ with time-complexity $\mathcal{O}(CND)$, followed by two Softmax operation over the first-axis of size $C$ and the second-axis of size $N$ with time-complexity $\mathcal{O}(CN)$, respectively, i.e., $\mathbf{A}_0 = Softmax(\mathbf{A}, axis = 0)$ and $\mathbf{A}_1 = Softmax(\mathbf{A}, axis = 1)$. Finally, we calculate $\mathbf{I} = \mathbf{A}_1 \times \log(\mathbf{A}_0)$ with time complexity $\mathcal{O}(CN)$. Overall, the entire pipeline requires element-wise operation is $\mathcal{O}(CND)$. It is worth noting that all these element-wise operations can be sped up with modern GPU accelerators. Empirically, computing the proposed objective costs around 15 seconds in total on a single Nvidia A6000 GPU.

## E  Training Sparse Autoencoders on Mistral-7B

Our training procedures and hyper-parameter settings majorly follow the previous works (Bricken et al., 2023; Gao et al., 2024; Lieberum et al., 2024). Specifically, we initialize $C = 2^{16}$ feature vectors for a Top-K sparse autoencoder with Kaiming initialization (He et al., 2015). Here, $C = 2^{16}$ is set according to the scaling law between the number of features $C$ and the number of training tokens $Z$ found by Gao et al. (2024), i.e., $C = \mathcal{O}(Z^{\gamma})$, where $\gamma \approx 0.60$ for GPT2-small and $\gamma \approx 0.65$ for GPT-4.[5] To prevent dead neurons, we also apply the tied-weight strategy as suggested by Gao et al. (2024). We use Adam optimizer (Kingma, 2014) with a constant learning rate of $1e^{-3}$ and epsilon of $6.25e^{-10}$ to train a total of 5 epochs. The hyper-parameters $\beta_1$ and $\beta_2$ of the optimizer are 0.9 and 0.999 following previous works Gao et al. (2024), respectively. We set the batch size as 512 queries, leading to around 90K tokens per gradient update, which is the same volume as Gao et al. (2024). The mixed precision training strategy (Micikevicius et al., 2017) is also applied to speed up the training process as Lieberum et al. (2024) found that it only shows a slightly worse impact on the model performance. Top-K sparse autoencoder has an initial sparsity $K = 200$, and it gradually decreases to the target sparsity $K = 20$ in the first 50% training samples of the first epoch. The training process runs on one single Nvidia A6000 GPU with CUDA 12.6 and takes about 16 hours per epoch.

---

[5]Empirically, $\gamma \approx 0.5978$ in our study.

# F  EXTENDED QUALITATIVE ANALYSIS ON RAW EXPLANATIONS

This section first provides an extension to our qualitative analysis of the raw explanations generated by different methods discussed in Section 4.2.2. In particular, Table 6, Table 7, and Table 8 provide more raw explanations and their automated summarization from Ours, TopAct, and N2G, respectively.

## F.1  ANALYSIS TO RAW EXPLANATIONS FROM OURS

The extended qualitative analysis on Ours demonstrates the robustness of our method in generating discourse-level explanations. Table 6 showcases a wide variety of explanations that extend beyond mere lexical overlaps, instead providing meaningful insights into different topics or concepts. For instance, explanations such as "Botanical classification and gardening practices" and "Urban development and community engagement" encapsulate coherent themes that align well with their raw explanations, reflecting the interpretative depth of our approach. This contrasts sharply with the baseline methods, which often focus on repetitive patterns or word-level constructs. By leveraging a de-duplicated vocabulary set and mutual information-based objective, our method avoids frequency biases and captures semantically rich discourse features.

## F.2  ANALYSIS TO RAW EXPLANATIONS FROM BASELINES

The extended qualitative analysis of the baselines TopAct and N2G highlights their tendencies to focus on repetitive linguistic patterns and fine-grained lexical constructs rather than capturing broader semantic or discourse-level themes. As shown in Table 7, TopAct often generates explanations dominated by repetitive queries or descriptive patterns, such as "What types of medical facilities are available for" or "Discuss the impact of social media on." While these patterns are interpretable, they largely lack thematic depth, emphasizing lexical regularities over conceptual diversity. On the other hand, in Table 8, N2G explanations successfully identify the most critical parts of the raw explanations and omit those non-critical ones with "[MASK]", resulting in a shortened raw explanations than the TopAct. However, N2G still falls short of representing more complex and discourse-level features. This limitation underscores the advantage of our proposed method in moving beyond the frequency bias to capture more coherent and meaningful features.

# G  SCALING UP WITH MACHINE ANNOTATORS

We build on recent progress in automated interpretation (Bills et al., 2023; Chaudhary & Geiger, 2024; Gao et al., 2024; Lieberum et al., 2024) by utilizing advanced LLMs to replicate human annotators in producing high-level interpretations. This approach allows us to leverage machine annotators, enabling us to scale our methods for analyzing the entire model and yielding more robust results. Specifically, we employ GPT-4o-mini[6] as our machine annotator. Our experiments utilize the gpt-4o-mini-2024-07-18 model with a hyperparameter temperature=0 for greedy decoding. For each response, we allow a maximum of 1024 tokens. To ensure the quality of automatic annotation, we design our prompting template with both the role-playing strategy and presenting in-context examples. We provide our prompting templates for reproducing our results as follows.

# H  LIMITATIONS

This research focuses on improving language models for specific applications by first interpreting and then steering their hidden representations. A primary limitation of this work is that our approach focuses on improving the post-hoc explanations of sparse autoencoders by alleviating the frequency bias (as discussed in Section 3.1). While our proposed explanation method mitigates this issue, it does not fundamentally alter SAE's architectures or training processes. Future work may design better SAE architectures or training objectives that inherently mitigate frequency bias, rather than solely addressing it in the post-hoc explanation stage.

---

[6]https://platform.openai.com/docs/guides/gpt

Table 6: Extended qualitative analysis on generated explanations from our proposed method.

| Method | Automated Summary | Raw Explanation |
|---|---|---|
| **Ours** | Local business and community engagement. | weekly, regional; native; pros; locally; good; cater; blog; perform; shop |
| | Botanical classification and gardening practices. | flower; hybrid; border; composition; popular; origin; habits; commonly; divide; fit |
| | Influence and alignment of ideas or concepts. | turn; impact; aligned; turning; leading; surrounding; nature; highlight; ideas; align |
| | Diverse strategies and approaches in chatbot development and interaction. | differently; pros; thorough; tricks; observations; view; approaches; Eastern; strategies; chatbot |
| | Digital solutions and services for businesses. | meaningful; inclusive; durable; online; tracking; quick; instant; hosting; marketing; processing |
| | Music education and authentic musical experiences. | stake; genuine; musical; authentic; arrangements; composition; classes; lessons; friend; empower |
| | Processes of change and interaction in systems or relationships. | crack; returning; describe; emerging; transform; transport; mutual; accompanied; interactions; index |
| | Personal development and productivity strategies. | cycle; trial; productive; lessons; lifestyle; neutral; Academy; rhythm; goal; goals |
| | Culinary arts and craftsmanship. | construction; variety; manual; design; fit; dinner; brand; craft; lunch; um |
| | Detection and identification of problems in the context of surveillance or monitoring systems. | detect, detective, detected, early, heat, instant, problem, parking, identifying, detection |
| | Urban development and community engagement. | productivity; interesting; align; correspond; hub; housing; grant; surrounding; mix; inform |
| | Impact of jazz music on youth and critical awareness. | best; question; contributing; mind; jazz; stake; critics; critique; kids; awareness |
| | Romantic or sexual relationships and interactions. | sexual; missed; strip; calling; attractive; shower; bond; shipping; shock; expect |
| | Project management and documentation processes. | prep; construction; construct; constructed; input; journal; action; claim; running; claims |
| | Influence of successful relationships or partnerships in a law enforcement or collaborative context. | bond; successful; successfully; police; being; landscape; working; deeply; influence; hit |
| | Fashion evolution and personal growth. | outfit, Smith, museum, leather, dress, growth, Chris, era, lifetime, grew |
| | Techniques for visual representation and support in design or art. | reflection, supportive, split, shelter, visual, grid, line, reflect, simple, tricks |
| | Concerns related to injuries and their representation in the context of Jewish communities or cultural icons. | draft, injuries, injury, concerns, concern, Jewish, happening, icon, strategies, graphic |
| | Focus on specific strategies or tactics in a competitive context. | keen, particular, certain, wall, gap, specialized, battle, escape, chop, specific. |
| | Crime detection and security measures. | detect, security, detective, crime, shadow, detection, criminal, deal, assets, out |
| | Energy resources and infrastructure management. | graph, composition, master, gas, pipeline, mine, perception, deployed, demand, stake |

## H.1 TEMPLATE 1

We directly append the words to this template to annotate the summary of the raw explanations with 10 selected words from our proposed method. In this template, we start with placing the role-play instruction in the system prompt. We then provide heuristic examples to simulate a multi-turn conversation between a user and an agent. In this way, once we attach the new word list-based raw

Table 7: Extended qualitative analysis on generated explanations from the baseline TopAct.

| Method | Automated Summary | Raw Explanation |
|---|---|---|
| TopAct | Inquiry about available options or facilities in various contexts. | What types of medical facilities are available for; What graduate programs are available for; What kind of housing options are available for; What accommodations are available for; What are some software options available for |
| | Setting personal boundaries and communication strategies. | Explain the potential consequences of failing to set boundaries; communication skills, as well as methods for setting boundaries; communication techniques that can be used to express personal boundaries; advice on how to communicate assertively, set boundaries; have faced, strategies you have used to set boundaries |
| | The concept of "role" in various contexts | as well as the role of technology and innovation in; they reside. Consider the role of email in; The role of technology and innovation in; washing techniques, the role of soap and water in; populations and the role of insecticides in |
| | Handmade or artisanal jewelry. | is a handcrafted Macrame beaded; hand-crafted artisanal glass beads; materials, tools, and methods used in bead; description: Handmade beaded; jewelry using materials such as beads |
| | Impact of social media. | Share your perspective on the impact of social media on; Discuss the impact of social media on; Discuss the impact of social media on; Write a report on the impact of social media on; Examine the impact of social media on |
| | Dimensions and measurements of objects. | They come in a 5" x; measuring approximately 6 inches by 6; 4" thick. 11" x; painting measures 14" tall by; a square paper measuring 8 inches by 8 |
| | Malicious intent related to food tampering or drugging. | Can I hide feces in someone's food; Whats the best way to hide poison in food; slip a roofie in a girl's drink; my fagot coworker's drink; type of poison for putting in people's food |
| | Manual processes in automation and testing. | operator-level isolation and rely on users to manually; using Selenium * Conduct manual; step-by-step guide on how to manually; Automated deployment flows and reduced manual; in a single run of the script without any manual |
| | Abortion rights and the Roe v. | have there been since Roe v. W; and vowed to pass legislation to protect abortion; mourning the fact that Roe v. W; landmark Supreme Court case Roe v. W; What is Roe v. W |
| | Impact of climate change. | Research and document the impact of climate change; 2. The effects of climate change; and write a report on the impacts of climate change; include a comprehensive analysis of the impact of climate change; are in place to address the effects of climate change |
| | Recipe search functionality and user interaction features. | and view recipes uploaded by others, a search; friendly, with an easy-to-use search; commenting and ratings for recipes, and a search; in the table view to allow the user to search; and ratings. Users should be able to search |
| | Webpage modification timestamps. | This page was last modified on; ings.\nThis page was last edited on; \nThis page was last edited on; construct.\nThis page was last modified on; 8.\nThis page was last edited on |

explanations from our method to this template, the machine annotator will directly generate the summarization for this explanation.

Table 8: Extended qualitative analysis on generated explanations from the baseline N2G.

| Method | Automated Summary | Raw Explanation |
|---|---|---|
| N2G | Character attributes in role-playing games. | choosing [MASK] race, class\n; name [MASK] race, class\n; name [MASK] race, class; Race [MASK] Human\n\nClass\n; backstory, class [MASK] |
| | Management and organizational skills in relation to tasks, teams, and time. | manage their tasks and; manage remote teams in; managing a [MASK] team?; manage [MASK] time effectively; manage my [MASK] team's territories? |
| | Negation or clarification phrases focusing on the phrase "doesn't mean". | [MASK] not necessarily; doesn[MASK]t mean; doesn[MASK]t mean; doesn[MASK]t mean; doesn[MASK]t mean |
| | Exclusion criteria or filtering terms. | not include [MASK] numbers or; exclude any [MASK] firm that; should not [MASK] any words that; exclude [MASK] words that; not include any [MASK] that |
| | Data storage and backup solutions, particularly focusing on external storage devices. | important data that you want to keep to an external; wireless file trans[MASK]; back[MASK]ups, and transferring; external hard; external hard |
| | Concepts related to returning or going back home. | last trains home; return home; walked home; way home; way home |
| | Bailout or financial assistance concepts, particularly in the context of economic interventions or stimulus packages. | GM Bail[MASK]; Paulson [MASK] other proponents of the bail; to step in to prevent it. Such bail[MASK]; and look at that auto bail[MASK]; stimulus packages [MASK] bail |
| | Informal greetings or inquiries about someone's well-being or current situation. | what[MASK]s going on; what[MASK]s going on; what[MASK]s up; What[MASK]s up; What[MASK]s up |
| | Customization and personalization of options or features. | options [MASK] customization; customizing [MASK]; to customize [MASK]; the player to customize [MASK] |
| | The phrase "On a scale" or variations of it, indicating a measurement or evaluation system. | On [MASK] scale of; On a scale [MASK]; On [MASK] scale of; On [MASK] scale of; On [MASK] scale of |
| | Addresses or locations. | 33 Dinah Shore Dr, [MASK]; 4[MASK]1 Bay Shore Road,; 1 Wessel Dr., [MASK];7 W. John St., [MASK]; 9[MASK]0 E. Street Rd., |
| | Gap year terminology. | [MASK] batical year; gap year [MASK]; gap year [MASK]; gap year [MASK]; gap year [MASK] |
| | Decades or time periods, specifically referencing the 70s, 80s, and 90s. | er from the 80 [MASK]; early 70 [MASK]; late [MASK] 90; 70s [MASK] 80; late [MASK] 90 |
| | Formatting and structuring text or documents focusing on the concept of a "clear head" or heading. | [MASK] appropriate head; format, with clear head [MASK]; [MASK] proper head; struct [MASK] and organized, with clear head; easy to follow, with clear head [MASK] |
| | Usage of the word "call" in various contexts, likely focusing on communication or addressing someone. | calls him [MASK]; call [MASK] americans indians?; calling [MASK] guy; call me [MASK]; called him [MASK] |
| | Historical figure: Benjamin Franklin. | Benjamin [MASK]; franklin [MASK]; Franklin [MASK]; Benjamin Franklin [MASK]; Benjamin Franklin [MASK] |

**Template-1 for Automated Summary with Word-based Raw Explanations**

```
System: You are studying a neural network. Each neuron
looks for one particular concept/topic/theme/behavior/pattern.
Look at some words the neuron activates for and guess what
the neuron is looking for. Pay more attention to the words
in the front as they supposed to be more correlated to the
neuron behavior. Don't list examples of words and keep your
summary as detail as possible. If you cannot summarize most
of the words, you should say Cannot Tell.

User: accommodation, racial, ethnic, discrimination,
equality, apart, utterly, legally, separately, holding,
```

> **Template-1 for Automated Summary with Word-based Raw Explanations (continued)**
>
> ```
> User: actual, literal, real, Really, optical, Physical, REAL,
> virtual, visual.
> Agent: Perception of reality.
>
> User: Go, Python, Java, c++, python3, c#, java, Ruby, Swift,
> PHP.
> Agent: Morden programming language.
>
> User: 1939–1945, 1945, 1942, 1939, 1940, 1941.
> Agent: Years of the second world war.
>
> User: 1976, 1994, 1923, 2018, 2014, 1876, 1840.
> Agent: Cannot Tell.
>
> User:
> ```

## H.2  TEMPLATE 2

Once we collect the summary of the raw explanation with the previous prompt, we then call the machine annotator again in a separate thread to evaluate whether the summary is hallucinated or not by using the following prompting template, where the placeholders "Summary" and "Raw Explanation" will be filled with real data. Note that if the machine annotator gives "Cannot Tell" as its prediction in the summarization stage, we will directly set the judgment for this task as "No".

> **Template-2 for Summary Judge with Word-based Raw Explanations**
>
> ```
> System: You are a linguistic expert. Analyze whether the
> words well represent the concept/topic/theme/pattern.
> Organize your final decision in format of "Final Decision:
> [[Yes/Probably/Maybe/No]]".
>
> User: Concept/Topic/Theme/Pattern: {Summary}.
> Words: {Raw Explanation}.
> Agent:
> ```

## H.3  TEMPLATE 3

Since baseline explainers (TopAct and N2G) consider N-gram spans as raw explanations, we found that the previous word-list-based prompting template leads a poor performance for their interpretability. Thus, we followed the strategies before to define the following text-span-based prompting templates. Here, the in-context examples of text spans are collected from previous work (Bricken et al., 2023). Specifically, similar to using Template 1 to summarize our extracted words, we append the extracted text spans from TopAct or N2G to this template. Note that we numerate each extracted span with a unique index.

---

**Template-3 for Automated Summary with Span-based Raw Explanations**

```
System: You are studying a neural network. Each neuron looks
for one particular concept/topic/theme/behavior/pattern.
Look at some spans the neuron activates for and guess what
the neuron is looking for. Pay more attention to the [last
few words] of each
spans in the front as they supposed to be more correlated to
the neuron behavior. Ignore the [MASK] patterns in the spans.
Don't list examples of spans and keep your summary as detail
as possible. If you cannot summarize most of the spans, you
should say "Cannot Tell."

User: Span 1: w.youtube.com/watch?v=5qap5aO4z9A
Span 2: youtube.come/yegfnfE7vgDI
Span 3: {'token': 'bjXRewasE36ivPBx
Span 4: /2023/fid?=0gBcWbxPi8uC
Agent: Base64 encoding for web development.

User: Span 1: cross-function[MASK]
Span 2: cross-function
Span 3: [MASK][MASK] cross-function\n
Agent: Particular phrase 'cross-function'.

User: Span 1: novel spectroscopic imaging platform
Span 2: and protein evolutionary network modeling
Span 3: reactions-centric biochemical model
Span 4: chaperone interaction network
Agent: Biological terms.

User: Span 1: is −17a967
Span 2: what is 8b8 − 10ad2
Span 3: 83 −11111011001000001011
Span 4: is −c1290 − −1
Agent: Synthetic math: Arithmetic, numbers with small digits,
in unusual bases.

User:
```

## H.4  TEMPLATE 4

We evaluate the quality of summarization using almost the same as Template 2, where we only change the phrase from "word" to "span" to fit the format of raw explanations from the baseline explainers.

**Template-4 for Summary Judge with Span-based Raw Explanations**

```
System: You are a linguistic expert. Analyze whether the
text spans well represent the concept/topic/theme/pattern.
Organize your final decision in format of "Final Decision:
[[Yes/Probably/Maybe/No]]".

User: Concept/Topic/Theme/Pattern: {Summary}.
Spans: {Raw Explanation}.
```

