# OpenReview forum: "Interpreting and Steering LLMs with Mutual Information-based Explanations on Sparse Autoencoders"
_ICLR.cc/2026/Conference — Submitted to ICLR 2026_

### Official Review · Reviewer_rrmv · 2025-10-26

**Soundness:** 3
**Presentation:** 3
**Contribution:** 2
**Rating:** 6
**Confidence:** 4

**Summary:**

The paper works around the challenge of interpreting and steering LLMs via SAEs. The authors claim "frequency bias" in current methods results in high-frequency linguistic patterns rather than deep semantic/discourse-level concepts. They propose a de-duplicated vocabulary and a mutual information-based objective for selecting explanations of SAE features. They also propose two strategies to steer LLMs using their explanations: amplification and calibration.

**Strengths:**

The paper is well-written and organized.

They clearly identify and support their claim regarding the frequency bias issues within LLMs.

They advocate their work by answering the following two questions:
RQ1: Does the proposed method generate more discourse-level explanations than traditional methods?
RQ2: Whether these discourse-level explanations useful in steering LLM behaviors?

The experiments are fairly comprehensive w.r.t model selection, datasets, and previous baselines.

**Weaknesses:**

The evaluation heavily relies on an LLM as a judge. It would be better and more reliable to have a human in the loop.

The assumption that $p(e_C = W_c) \approx 1$ is not justified. I feel like assuming that $W_c$ represents a unique topic $e_C$ is a strong assumption.

**Questions:**

Have you done any ablation where only one of the de-duplication or the MI-based is considered? I'm wondering if vocabulary pruning is more effective or the new objective?

Is the proposed method generalizable to other applications, rather than jailbreak? For instance, for style control or factuality enhancement.

---

### Official Review · Reviewer_tgAp · 2025-10-28

**Soundness:** 2
**Presentation:** 2
**Contribution:** 2
**Rating:** 2
**Confidence:** 3

**Summary:**

This paper points out a common issue when interpreting SAE features, which is that using top-activating tokens from the corpora yields repetitive phrases. They present a new method to extract top tokens that are less repetitive, which can then be used to label SAE features. They do a steering experiment with these new feature descriptions.

**Strengths:**

The authors show  clear evidence that prior methods to interpret SAE features might be dominated by repetitive text strings in training corpora.

**Weaknesses:**

1. Clarity of some sections could be improved. For example, could the authors more clearly describe the metrics they use to evaluate how good an explanation is - like the Explained Rate (4.2)?
2. **Steering experiment?** Please clarify the steering experiment, as is it's not really clear how you are comparing your explanation method to the authors in this experiment.
3. **Tasks beyond steering:** The paper would be stronger if there were a clearer link to how the proposed explanation method will help in downstream tasks besides steering, which recent work finds that SAEs lose to baselines on [4]. For example, does the proposed method improve hypothesis generation [1] or understanding of model biology [2]?
4. It isn't clear if the proposed explanation method actually yields natural language descriptions that better predict the SAE feature activations compared to baselines. For example, some measure of explanation fidelity or generation scoring [3] would be useful. If this is how the "Explained Rate" metric already works, the authors should clarify this.


[1] https://arxiv.org/abs/2502.04382
[2] https://transformer-circuits.pub/2025/attribution-graphs/biology.html
[3] https://blog.eleuther.ai/autointerp/
[4] https://arxiv.org/abs/2501.17148

**Questions:**

1. It's hard to compare this method against the baselines (e.g. in Table 1) because all the methods are interpreting different features. Can we see the *same* features and the top spans that different methods extract?
2. How does the experiment in 4.3.2 work (details are currently unclear), and how does it show that the authors' proposed explanation method is better?
3. One simpler approach to mitigate the issue of repetitive phrases is just to use TopAct, except with a diversity filtering step. Did the authors try this? (e.g. sample from the top 10% bin of positive activations, and exclude any examples with very high cosine similarity with others)

---

### Official Review · Reviewer_tD4k · 2025-10-31

**Soundness:** 3
**Presentation:** 3
**Contribution:** 1
**Rating:** 4
**Confidence:** 3

**Summary:**

The paper studies post-hoc explanations for sparse autoencoder (SAE) features in LLMs and argues that prior explainers overfit to frequent linguistic patterns instead of rarer discourse topics (“frequency bias”). It proposes: (1) explaining features with a de-duplicated vocabulary and a mutual-information (MI) objective that normalizes a word’s affinity across all features; and (2) two runtime steering methods—Amplification and Calibration—that adjust selected feature activations. Experiments (Mistral-7B-Instruct; additional Gemma-2 in the appendix) show higher explanation quality (more discourse-level, less frequency bias) compared to TopAct/N2G/LogitLens and improved jailbreak defense (lower ASR on Salad-Bench), with a small, yet helpful, impact on MT-Bench.

**Strengths:**

1. The topic-model-based analysis motivates why discourse features are rarer and how this induces frequency bias in SAE explanations.

2. Amplification and Calibration are simple residual-space edits, targeting only a small feature subset at runtime.

3. Steering with safety-related features lowers jailbreak ASR (81.6→72.8) with modest MT-Bench degradation (6.5→6.0) and little/no latency overhead compared to prompting or perturbation defenses.

**Weaknesses:**

1. The contribution of the paper is valid but incremental. The overall contribution is to enhance the SAE-based explanation in response to a frequency bias. The paper identifies the bias, provides two methods to mitigate it, and offers implications for safety. This makes the paper an ensemble of many small or trivial contributions, with each contribution being shallow rather than deep. I would suggest the paper be more focused on 1 or 2 points (e.g., deeper analyses on why frequency bias happens) instead of a series of interconnected small points

2. Explanation quality relies on LLM-as-judge summaries/labels; human or task-grounded evaluations would strengthen claims.

3. Core results use Mistral-7B-Instruct with SAEs trained on ~113M tokens from ~711k prompts; broader model scales, layers, and domains are only lightly explored.

**Questions:**

1. Can the paper emphasize/clarify 1 or 2 contributions that are deep?

2. What is the wall-clock and memory overhead of monitoring |S| features for Amplification/Calibration across long contexts and larger batches?

3. How do MI explanations vary with vocabulary size M, feature count C, and layer choice? Is there any principled guidance beyond the 8th-layer focus?

4. Can you include human-rated coherence of explanations and human-rated utility/safety on a subset to corroborate LLM-as-judge results?

5. Beyond Mistral/Gemma, do results hold for instruction-tuned Llama-3 or command-style models, and for non-safety steering (e.g., style or reasoning control)?

6. Since the paper also mentions safety, how robust are MI explanations/steering under distribution shift, adversarial paraphrases, or multilingual prompts?

---

### Meta-Review · Area_Chair_ZVTm · 2026-01-04

**Summary:**

There were substantial concerns regarding the depth of contribution, the rigor of evaluation, and the strength of empirical evidence connecting improved explanations to broader interpretability or control benefits.

**Reviewer Concerns:**

Limited depth / incremental contribution.
Multiple reviewers found the paper to be an “ensemble of small ideas,” where each component (de-duplication, MI scoring, and runtime editing) is conceptually straightforward and does not offer a sufficiently deep or principled advance over existing SAE interpretation pipelines.

Evaluation relies heavily on LLM-as-judge and indirect proxies.
The primary evidence for “better explanations” depends largely on automatic/LLM-based judgment, and there is limited human or task-grounded validation. Moreover, it remains unclear whether the proposed explanations improve fidelity—i.e., whether they better predict or characterize when a feature activates—beyond producing more diverse or discourse-like labels.

Unclear causal link between improved explanations and downstream utility.

Comparisons and baselines are not fully convincing.

**Reviewer Scores:**

There is no rebuttal posted. The weaknesses mentioned by the reviewers still hold.

---

### Decision · Program_Chairs · 2026-01-26

Reject